# Confirmatory factor analysis and exploratory structural equation modelling of the factor structure of the Depression Anxiety and Stress Scales-21

**Rapson Gomez[1], Vasileios Stavropoulos[2]\*, Mark D. Griffiths[3]**

**1** Federation University, Ballarat, Australia, **2** Victoria University, Melbourne, Australia, **3** Nottingham Trent University, Nottingham, England, United Kingdom

\* Vasilisstavropoylos80@gmail.com

**Data Availability Statement:** All relevant data are within the manuscript and its Supporting Information file.

## Abstract

The Depression Anxiety and Stress Scales-21 (DASS-21) involves a simple structure first-order three-factor oblique model, with factors for depression, anxiety, and stress. Recently, concerns have been raised over the value of using confirmatory factor analysis (CFA) for studying the factor structure of scales in general. However, such concerns can be circumvented using exploratory structural equation modeling (ESEM). Consequently, the present study used CFA and ESEM with target rotation to examine the factor structure of the DASS-21 among an adult community. It compared first-order CFA, ESEM with target rotation, bi-factor CFA (BCFA), and bi-factor BESEM with target rotation models with group/specific factors for depression, anxiety, and stress. A total of 738 adults (males = 374, and females = 364; M = 25.29 years; SD = 7.61 years) completed the DASS-21. While all models examined showed good global fit values, one or more of the group/specific factors in the BCFA, ESEM with target rotation and BESEM with target rotation models were poorly defined. As the first-order CFA model was most parsimonious, with well-defined factors that were also supported in terms of their reliabilities and validities, this model was selected as the preferred DASS-21 model. The implications of the findings for use and revision of the DASS-21 are discussed.

## Confirmatory factor analysis and exploratory structural equation modeling of the structure of attention-deficit/hyperactivity disorder symptoms

The Depression Anxiety Stress Scales-21 (DASS-21) is a widely used questionnaires. It has three separate, but correlated, scales (with seven items for each scale) for depression (assessing dysphoria, low self-esteem, and lack of incentive), anxiety (assessing somatic and subjective responses to anxiety and fear), and stress (assessing negative affectivity responses, such as nervous tension and irritability). Thus, the theoretically proposed factor model for the DASS-21 is

**Funding:** The author(s) received no specific funding for this work.

**Competing interests:** The authors have declared that no competing interests exist.

a simple structure first-order three-factor oblique model, with factors for depression, anxiety, and stress. Asparouhov and Muthén [1] (see also [2]) have developed an advanced approach for testing the factor structure of a measure called exploratory structural equation modeling (ESEM) with target rotation. To date, this technique has been applied to valuate the factor structure of DASS-21 in a community sample. Consequently, the major aim of this research was to use this approach to examine the factor structure of the DASS-21 in adults from the general community.

### Initial development and validation of the DASS-21

In the initial development and validation study of the DASS-21, based on students, Lovibond and Lovibond [3] suggested sufficient reliability, convergent and discriminant validity. Calculations suggested that all items associated to their allocated three dimensions, with rates interrelating moderately. Using the same population, Lovibond and Lovibond [3] supported a better fit for a three-factor oblique structure than a uni-dimensional (all items associating to a single dimension) or a two-dimensional oblique structure (depression items associating on one dimension, and anxiety and stress items associating on another dimension). Therefore, the DASS-21 has been conceptualized as a three-dimensional oblique structure, involving dimensions for depression, anxiety and stress ([3]; see Fig 1). In this model, all the different depression, anxiety, and stress items only associate to their allocated latent dimensions (simple structure). Also, the latent dimensions are correlated with each other, and there are no cross-loadings and all error variances are freely estimated and all covariances between error variances are set to zero. Although there have been some minor misspecifications in terms of cross-loadings, other PCA and exploratory factor analysis (EFA) studies of the DASS-21 have generally supported the theorized three-factor model (e.g., [4–9]).

### Confirmatory factor analysis of the DASS-21

In addition to PCA and EFA, CFA has also been used to test the DASS-21 dimensionality. Methodologically, EFA procedures allow unlimited item-factor cross-loadings. Therefore, items are enabled to associate with various dimensions concurrently. In contrast, CFA, which follows a model-based process, allows the researcher to test for *a priori* defined factor structure. Thus, in this approach (more specifically the ICM-CFA approach), items only load on their *a priori* designated target factors, with no cross-loadings [10,11]. A bi-factor model or a BCFA model is also an *a priori* CFA model. In this model, there is one general factor and two or more specific factors. Typically, all the items in the measure load on the general latent, and also items belonging to each group load on their own specific latent factors, and all the latent factors are not correlated with each other (orthogonal model). Thus, the overall general dimension indicates the common (shared) variances between all the items in the measure and the specific dimensions reflect variances in them that are not accounted for by the general factor.

Since the publication of DASS-21, numerous CFA studies have supported the theorized three-factor model [8,12,13,14]. More recently, many studies have examined the structure of DASS-21 in terms of a BCFA model with three-specific factors, namely depression, anxiety, and stress [15,16,17]. Fig 1 also includes the path diagram for the DASS-21 BCFA model with the three specific factors. In brief, it has one general distress latent factor on which all the DASS-21 items load, a specific factor for depression on which only the depression items load, a specific factor for anxiety on which only the anxiety items load, and a specific factor for stress on which only the stress items load. In this model, the overall and specific dimensions are not correlated (orthogonal model). Consequently, the overall dimension reflects the shared

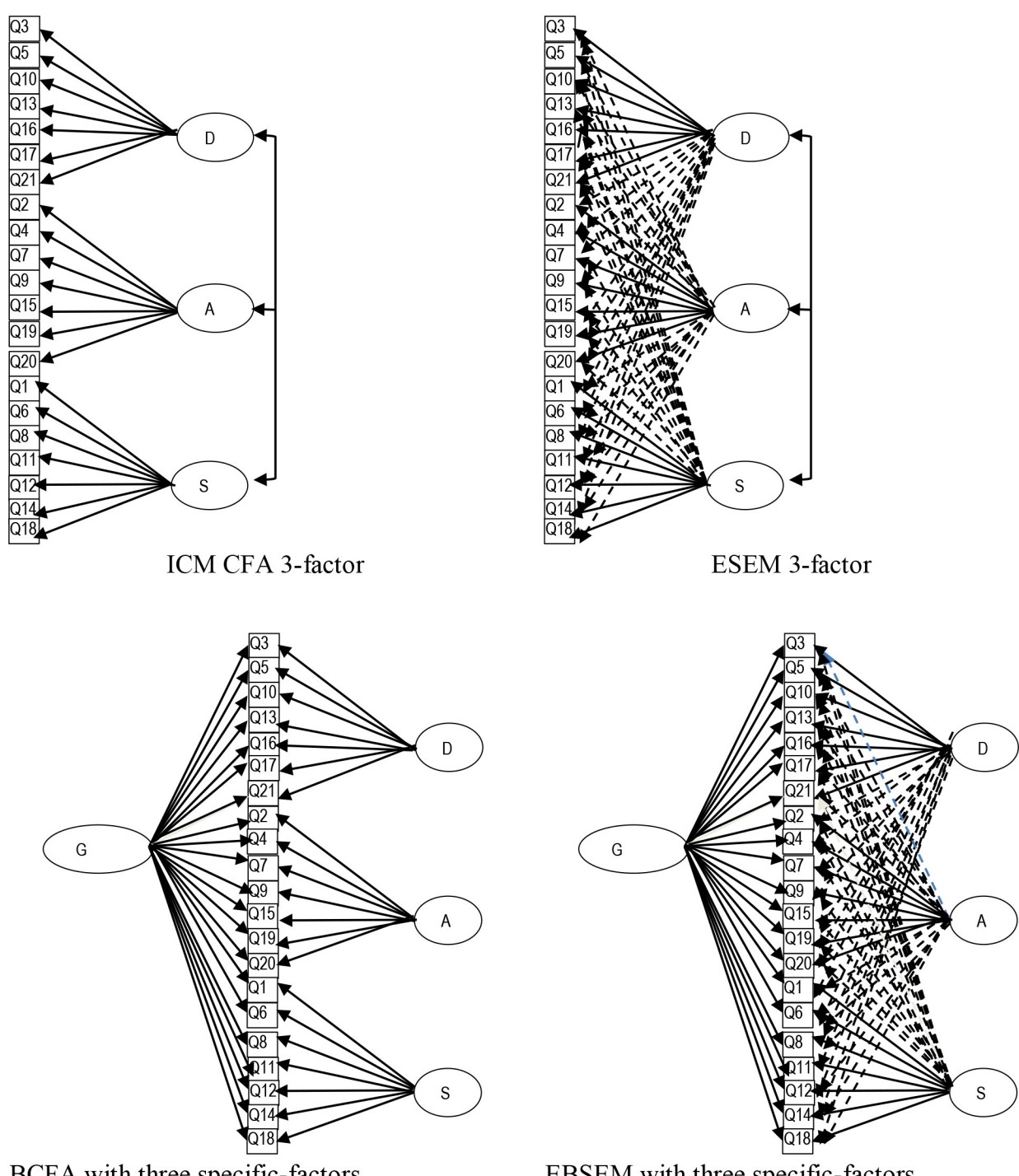

ICM CFA 3-factor

ESEM 3-factor

BCFA with three specific-factors

EBSEM with three specific-factors

**Fig 1. Conceptual representation of CFA, ESEM, BCFA, and BESEM models.** D = depression; A = anxiety; S = stress; G = general factor; ESEM = exploratory structural equation modeling.

variances between all the 21 DASS-21 items and the specific dimensions reflect variances in them that are not accounted for by the general factor. Generally, the findings in many of the BCFA studies, cited earlier, have supported this model, with it (when examined) having better

global fit than the three-factor CFA model. Despite the fact that the BCFA model has shown better global fit for the DASS-21 than the CFA model, it is uncertain what underlies this finding. Related to this, there have been recent criticisms over the application of CFA and BCFA approaches for testing the factor structure of complex measures, such as the DASS-21.

## Limitations of CFA and bi-factor approaches

The restriction in CFA that items are associate exclusively to their designated dimensions, and all the loadings on non-designated dimensions (cross-loadings) are limited to zero [10,11] is viewed as highly compromising because items are infrequently exclusive indicators of their designated dimensions [18]. Indeed, cross-loadings for DASS-21 items have often been observed in EFA studies of the DASS-21 [6,8,9]. For example, the study by Oei et al. [7] found significant cross-loadings for seven of the DASS-21 items. Therefore, CFA may not accurately reflect the dataset, often resulting to poor fit and could consequently show poor fit even when this is not the case [2,19]. In relation to BCFA, Bonifay, Lane, and Reise[2] [20] have pointed out other concerns. One concern is the difficulty in the interpretation of the meaning of specific factors in a bi-factor model. This is because statistically, the specific factors are seen as 'nuisance' variables, whereas from a theoretical viewpoint, the specific factors constitute substantially meaning factors that are not accounted for by the general factor. According to Park et al. [21], the appropriate explanation will depend on the reliability, stability, and incrementally validities of the specific factors. Another concern relates to the fact that the bi-factor model will always fit better than the corresponding first-order factor model because they can better accommodate nonsense response patterns in the dataset, thereby making it appear better even if this is not actually the case. A third concern is that because more parameters are estimated in a bi-factor model, this is a less parsimonious and potentially an unnecessary model compared to the corresponding CFA model.

To overcome the limitation of the CFA approach (i.e., all cross-loadings constrained to zero), the exploratory structural equation modeling (ESEM) with target rotation procedure has been developed [1,2]. ESEM with target rotation is a synergy of the EFA and CFA approaches, incorporating the advantages of the EFA approach (allowing cross-loadings) and CFA approach (model-based, and testing *a priori* defined structure). Fig 1 shows the ESEM with target rotation model as applied to DASS-21. As shown, the targeted symptoms load on their own factors as well as non-targeted factors (at values close to zero). Available evidence has demonstrated the advantages of the ESEM with target rotation over the EFA and CFA processes [2,19]. The basic ESEM with target rotation model can be expanded to a bi-factor ESEM (BESEM) with target rotation model, thereby combining concurrently the advantage of the CFA approach (*a priori* model specifications), the EFA approach (fallibility of the items and allowing cross-loadings), and the bi-factor approach (allowing a general factor and specific factors that are not correlated) [18,19]. Therefore, compared to the EFA, CFA, and ESEM with target rotation approaches, the BESEM approach with target rotation offers a more sophisticated method to explore different sources of construct-relevant multidimensionality [18]. This is especially relevant to the DASS-21 items because they have demonstrated multidimensionality, cross-loadings, and the presence of general and specific factors. Fig 1 includes the BESEM model with target rotation, as applied to the DASS-21 items, with specific factors for depression, anxiety, and stress. Additionally, for a bi-factor model, it is also important to evaluate if the factors in a bi-factor model are substantially meaningful. Morin et al. [18] have proposed an integrated test of multidimensionality procedure for this purpose. Additionally, Park et al. [21] have suggested evaluation of the reliabilities and validities of the factors in this model, especially the incremental validities of the specific factors.

As far as it can be ascertained by the present authors, the ESEM with target rotation approach has been applied in at least two studies involving the DASS-21 [6,7]. Johnson [6] examined the three-factor CFA, ESEM with target rotation, and BCFA models with three group factors (depression, anxiety, and stress) for ratings provided by individuals with Parkinson's disease. Although the findings indicated good fit for the ESEM with target rotation model, the anxiety factor was poorly defined, with several items loading significantly onto the depression or stress factors. Kyriazos [7] used CFA, ESEM with target rotation, and BCFA to examine the factor structure of the DASS-21 (with group factors for depression, anxiety, and stress) among an adult community. Although there was good support for all three models, the authors concluded most support for the CFA three-factor model. The primary reason for this was that in the BCFA and ESEM with target rotation models, the factor loadings were unsatisfactory (low and/or negative item loadings on targeted factors). However, because this study did not use BESEM with target rotation, which as has been argued, could be potentially a more sophisticated and arguably more advanced approach for studying the factor structure of the DASS-21, the support claimed for the CFA model needs to be viewed cautiously. In this respect, the study by Johnson [6] that involved individuals with Parkinson's disease used BESEM with target rotation to examine the structure of the DASS-21. Two different bi-factor models were examined: Henry and Crawford's bifactor Model (specific factors for depression, anxiety, and stress and a general factor) [22]; and Tully, Zajac, and Venning's bifactor model (specific factors for depression and stress, and a general factor) [23]. Although the BESEM with target rotation, based on Henry and Crawford's bifactor model showed slightly better fit than their eventually preferred ESEM with target rotation model, the authors concluded that the improvement was not sufficient to justify the loss of parsimony [22]. However, as this study examined individuals with Parkinson's disease, it cannot be assumed this this will also be the case for individuals from the general community. Consequently, further studies involving community samples are needed.

### Aims of the present study

Given the aforementioned limitations and omissions, the major aim of the present study was to examine the structure of the DASS-21 items among adults from the general community using the BESEM with target rotation approach. For this, the integrated test of multidimensionality procedure proposed by Morin et al. [18] was closely adhered to. Consequently, the present study also tested the CFA three-factor model, the ESEM three-factor model with target rotation, the BCFA model with three specific factors, and the BESEM with three specific factors target rotation model. The three factors in all models were depression, anxiety, and stress. These models were described earlier (see also Fig 1). For all four models tested, model-based reliabilities and support for the external validities of the different factors in them were also examined. In terms of predictions, most support for the BESEM with three specific factors with target rotation model in terms of global fit was expected, but it was also speculated that one or more of its specific factors may not be well defined.

## Method

### Participants

The sample comprised 738 adult individuals, from the general community, with ages ranging between 18 and 72 years (M = 25.29 years; SD = 7.61 years). There were 374 males (50.7%; M = 25.28 years, SD = 7.71 years), and 364 females (49.3%; M = 25.30 years, SD = 7.51 years). Table 1 shows additional background information. As shown in the table, more than half the participants were employed, and had completed technical or university education, Also, on

**Table 1. Frequencies and descriptive statistics of background variables, and descriptive for the DASS items.**

| Background variables | Frequency/ Descriptive Statistics |
|---|---|
| Employed | Frequency (percentage) = 529 (71.7%) |
| Highest Educational Level | |
| Primary | Frequency (percentage) = 42 (5.7%) |
| Secondary | Frequency (percentage) = 268 (36.3%) |
| Technical | Frequency (percentage) = 193 (26.2%) |
| University | Frequency (percentage) = 235 (31.8%) |
| Years of education | Mean (SD) = 12.01 (5.12) |
| ADHD—Inattention | Mean (SD) = 14.43 (6.20) |
| ADHD—Hyperactivity/Impulsivity | Mean (SD) = 14.28 (6.09) |
| DASS-21—Stress | Mean (SD) = 6.86 (4.34) |
| DASS-21—Anxiety | Mean (SD) = 5.36 (4.29) |
| DASS-21—Depression | Mean (SD) = 7.41 (5.65) |

average, participants completed at least 12 years of education. In relation to the DASS-21 items, their mean scores ranged from 0.54 to 1.42. On a scale of 0 ("*did not apply to me at all*") to 3 ("*applied to me very much or most of the time*"), these scores indicate low levels for all the items.

Based on the normative scores for the DASS-21, published by Henry and Crawford [22], the scores for the participants in the present study for depression, anxiety, and stress (see Table 1) were less than 1 *SD* from the mean, around 1.5 *SD* from the mean, and just above 1 *SD* from the mean, respectively. According to Kessler et al. [24], a total score of 24 or more for IA or HI symptom groups on the Adult ADHD Self-Report Scale Symptom Checklist (ASRS) is indicative of a high likelihood of an ADHD (attention deficit hyperactivity disorder) diagnosis. In the present study, the total mean scores for IA and HI were 14.43 and 14.28 respectively (see Table 1). Therefore, on the whole, the participants in the present study can be seen as reasonably well adjusted, with no problematic levels of depression, anxiety, stress, or ADHD symptoms. All relevant data are within the manuscript and its Supporting Information file" has now been added to the cover letter and the article's accompanying information.

## Measures

**Depression Anxiety Stress Scales-1 (DASS-21; [3]).** As noted above, the DASS-21 constitutes a self-addressed instrument involving 21 items, subdivided into three equal-size subscales reflecting depression, anxiety, and stress. Items are rated on a 4-point Likert (0 = did not apply-3 = applied most of the time) sequence in terms of how often the individual experiences the behavior they refer to during the past week. Past evidence has favored acceptable convergent and discriminant validities, and high internal reliabilities for the three DASS-21 subdimensions [3, Norton, 2007]. The internal reliability for the depression, anxiety, and stress dimensions in the present sample were .84, .71 and .83 respectively.

## Procedure

The study was approved by the Cairnmillar Institute Human Research Ethics Committee. Participants were recruited online via advertisements on various platforms, including social media (e.g., *Facebook*) and internet chatrooms (e.g., *Discord*). Individuals who clicked the survey link were debriefed on the first page with a description of the study and its aims via the Plain Language Information Statement. Verification that participants' data would be recorded

anonymously was provided, and a statement ensuring that they had the choice to stop participating in the survey at any point in time was also included. Participants digitally provided their informed consent by clicking to proceed to take part in the survey. Apart from age being adults, no other exclusion criteria were applied. Participants were not compensated in any way for their participation, or disadvantage for withdrawing from the study.

### Statistical analysis

All statistical calculations were implemented using M*plus* Version 7.3 and R [25,26]. The weighted least square mean and variance adjusted (WLSMV) estimator was applied. WLSMV can correct for non-normality in the dataset and is suited for responses with four or less response categories [26,27,28], as is the case with the items listed in the DASS-21.

To establish the best DASS-21 model, the integrated test of multidimensionality procedure proposed by Morin et al. [18] was closely followed. Therefore, in the initial phase, the global fit of the all the models tested was examined and compared. In a subsequent phase, the correlations between the factors in the CFA and ESEM with target rotation models were compared. The next phase involved evaluating the presence of construct-relevant psychometric multidimensionality due to the fallible nature of the indicators by comparing the factor loadings in the CFA and ESEM with target rotation models. In the next phase, the presence of a hierarchically superior latent construct was evaluated by comparing the fit of ESEM with target rotation and the BESEM with target rotation models, and then the patterns of loadings and crossloadings in the BESEM with target rotation model.

Because $\chi^2$ values, such as WLSMV$\chi^2$, are sample dependent, the fit here was additionally evaluated by approximate fit indices. These involved the root mean squared error of approximation (RMSEA), the comparative fit index (CFI), and the Tucker Lewis Index (TLI). Hu and Bentler [29] supported RMSEA rates approximating .06, or lower as good fit, 0.07 to 0.08 as acceptable fit, 0.08 to .10 as limited fit, and >.10 as unfit. Considering CFI and TLI, rates approximating .95 or higher were indicative of sufficient fit, between .90 and .95 as acceptable fit, and below .90 as poor fit. As the DASS-21 models assessed were nested, the chi-square difference test ($\Delta\chi^2$) was employed for examining differences in model fit. However, because the $\Delta\chi^2$ test is also highly sensitive to large sample sizes, for the present study, models were also compared using change in the RMSEA and CFI values, Generally, differences in CFI values of 0.01 or more and/or RMSEA values of 0.015 or more are interpreted as difference for the models being compared [30,31]. For all models tested, model-based reliabilities for the different factors were computed. More specifically, the categorical omega ($\omega$) values for the factors were computed alongside their explained Explained Common Variance (ECV) [32,33]. The ECV in the general factor of a bi-factor model reflects the degree of uni-dimensionality of the indicators, with higher values indicating more support for uni-dimensionality. ECV values > .70 for the general factor indicate that the factor loadings on this factor are close to that expected for a one-factor model [32,33]. The $\omega$ should be viewed as an index describing the amount of variance in summed (standardized) scores related to the specific dimension [34]. For a bi-factor model, the $\omega$ values for the general and specific factors are referred to as omega hierarchical ($\omega_h$) and omega-subscale ($\omega_s$) respectively [32,34]. As our data was categorical, we computed categorical omega [33,35,36,37]. The confidence intervals for categorical omega were computed using the "*ci.reliability*" function in the MBESS R package version 4.4.0 [33]. The values for all types of $\omega$ values fluctuate from 0 to 1, with 0 revealing no reliability and 1 revealing perfect reliability [33]. According to Reise, Bonifay, and Haviland [38], in a bi-factor model, $\omega_h$ values need to be at least .50 with values of at least .75 being preferable.

The external validity of the factors in all models tested were also examined. To test for the validity of the DASS-21 factors, the ADHD IA and HI total scales scores were regressed on the relevant factors in the DASS-21 models. To control for possible confounding effects of age and gender, both these variables were entered as covariates in the analyses. Significant and positive predictions of either IA or HI total score by a DASS-21 factor can be taken as support for the validity of that factor, In this context, significant and positive prediction by the general factor can be interpreted as supportive of the validity of that factor, and significant and positive predictions of IA or HI total score by the specific/group factors (depression, anxiety, and stress) can be taken as support for the incremental validity of the relevant specific factors.

## Results

There was no missing value in the data set.

### Comparison of global fit

Table 2 shows the fit values for all the DASS-21 models tested in the study. For all models, their CFI, TLI and RMSEA values indicated good fit. At the statistical level, CFA 3-F model differed from the ESEM 3-F ($\Delta df$ = 36; $\Delta WLSMV\chi^2$ = 152.04, $p < .001$), BCFA 3-s-F ($\Delta df$ = 186; $\Delta WLSMV\chi^2$ = 80.31, $p < .001$), and BESEM 3-s-F ($\Delta df$ = 186; $\Delta WLSMV\chi^2$ = 218.82, $p < .001$) models. The BCFA 3-s-F model differed from the ESEM 3-F ($\Delta df$ = 18; $\Delta WLSMV\chi^2$ = 84.32, $p < .001$) and BESEM 3-s-F ($\Delta df$ = 36; $\Delta WLSMV\chi^2$ = 152.30, $p < .001$) model. The ESEM 3-F model differed significantly from the BESEM 3-s-F ($\Delta df$ = 18; $\Delta WLSMV\chi^2$ = 74.11, $p < .001$) model. This means that the fit values for all models differed significantly from each other, with the BESEM 3-s-F model showing the best fit, followed in sequence by ESEM 3-F, BCFA 3-s-F, and ESEM 3-F models. In relation to approximate fit indices, the models did not differ from each other in terms of $\Delta$RMSEA values ($<$ O.015). However, the $\Delta$CFI was greater than 0.01 for the comparison involving the CFA 3-F with the ESEM 3-F and the BESEM 3-s-F models. There was no difference in the CFI values between the CFA 3-F and the BCFA 3-s-F models, or between the ESEM 3-F and the BESEM 3-s-F models. Taken together, the findings can be interpreted as showing that both the ESEM models (ESEM3-F and BESEM 3-s-F) showed better fit that the both the first-order models (CFA3-F and BCFA 3-s-F).

### CFA versus ESEM models

**Correlations between latent factors.**   Table 2 also includes the correlations of the latent factors in the CFA and ESEM with target rotation models. As shown, the correlations of the

**Table 2. Fit of all the models tested in the study.**

| Models | $\chi^2$ (*df*) | Fit Values | | | Factor correlation | | |
| --- | --- | --- | --- | --- | --- | --- | --- |
| | | CFI | TLI | RMSEA (90% CI) | D-A | D-S | A-S |
| CFA 3-F | 515.31 (186) | . 969 | .965 | . 049 (.044 - .054) | .73 | .79 | .90 |
| ESEM 3-F | 357.79 (150) | . 980 | .973 | . 043 (.038 - .049) | .59 | .53 | .45 |
| BCFA 3-s-F | 438.08 (168) | . 975 | .968 | 047 (.041 - .052) | | | |
| BESEM 3-s-F | 281.99 (132) | . 986 | .978 | . 039 (.033 - .046) | | | |

F = factor; D = depression; A = anxiety; S = stress; CI = confidence interval; $\chi^2$ = chi-square; *df* = degrees of freedom; CFA = confirmatory factor analysis; ESEM = exploratory structural equation modeling; BCFA = bi-factor confirmatory factor analysis; BESEM = bi-factor exploratory structural equation modeling; s = specific; RMSEA = root mean square error of approximation; CFI = comparative fit index; TLI = Tucker-Lewis Index.

latent factors were much lower in the ESEM with target rotation model than the CFA model. Generally, when this is the case, the ESEM with target rotation model is considered a better model than the CFA model to reflect the variances in the items [2,11].

**Factor loadings.** Table 3 shows the factor loadings for the CFA and ESEM with target rotation models. As shown, for the CFA model, all 21 items loaded significantly and saliently (> .30) on their targeted (designated) factors. For the ESEM with target rotation model, all seven depression items loaded on their targeted depression factor, and the loadings for all seven items were salient. Four anxiety items (Items 9, 15, 19 and 20) also had significant loadings on the depression factor, with the loading for one of them (Item 19) being negative. Also, with the exception of one item (Item 11), the other six stress items loaded significantly on the depression factor, with the loadings for three of them being positive and salient (Items 6, 8, and 12), one (Item 11) being close to salience, and one (Item 1) being salient and negative. In relation to the anxiety items, all seven items loaded significantly on their targeted anxiety factor. Of these, with the exception of one of them (Item 2), all the other items has salient and positive loadings. The loading for Item 2 was negative and not salient. Four depression items (Items 3, 13, 16 and 17) loaded significantly on the anxiety factor, with the loadings for two of them (Items 3 and 16) being negative. With the exception of one item (Item 14), the other six stress items loaded significantly on the anxiety factor, with three of them (Items 6, 8 and 12) also being salient and positive, one (Item 11) close to salience and positive, and one salient and negative (Item 1). In relation to the stress items, all seven of them had significant loadings on

**Table 3. Factor loadings of the first-order and bi-factor CFA and ESEM models.**

| # | Items<br>Brief description | CFA three-factor | | | ESEM three-factor | | | Bi-factor CFA three-factor | | | | BESEM three-factor | | | |
|---|---|---|---|---|---|---|---|---|---|---|---|---|---|---|---|
| | | D | A | S | D | A | S | G | D | A | S | G | D | A | S |
| 3 | Experiencing positive feelings | .74*** | | | .74*** | -.21*** | .24*** | .58*** | .48*** | | | .64*** | .44*** | -.19* | -.21* |
| 5 | Working up initiative | .65*** | | | .44*** | .09 | .19*** | .58*** | .23*** | | | .56*** | .24*** | -.03 | .11* |
| 10 | Nothing to look forward to | .80*** | | | .80*** | .03 | -.03 | .63*** | .49*** | | | .61*** | .52*** | .04 | .10** |
| 13 | Felt down-hearted | .80*** | | | .74*** | .09* | -.02 | .64*** | .45*** | | | .63*** | .47*** | .09* | -.01 |
| 16 | Unable to become enthusiastic | .77*** | | | .75*** | -.11* | .16** | .60*** | .48*** | | | .63*** | .45*** | -.11* | -.02 |
| 17 | Not worth much | .81*** | | | .80*** | .09* | -.08 | .64*** | .49*** | | | .60*** | .53*** | .10* | .11** |
| 21 | Life was meaningless | .81*** | | | .92*** | .00 | -.15*** | .60*** | .59*** | | | .57*** | .62*** | .08 | .07 |
| 2 | Dryness of mouth | | .40*** | | .08 | -.14* | .58*** | .39*** | | -.01 | | .47*** | -.06 | -.32** | -.11 |
| 4 | Difficulty breathing | | .65*** | | -.04 | .46*** | .36*** | .56*** | | -.11*** | | .66*** | -.14*** | .12 | -.07 |
| 7 | Trembling | | .52*** | | .09 | .38*** | .12* | .48*** | | -.05*** | | .50*** | .00 | .18* | -.03 |
| 9 | Worry about embarrassing self | | .71*** | | .20*** | .58*** | .02 | .67*** | | -.03* | | .62*** | .08* | .33*** | .12 |
| 15 | Felt close to panic | | .81*** | | .13*** | .66*** | .12* | .75*** | | -.05*** | | .74*** | .01 | .34*** | .05 |
| 19 | Awareness of action of heart | | .58*** | | -.10* | .47*** | .34*** | .50*** | | -.11*** | | .61*** | -.18*** | .13 | -.05 |
| 20 | Felt scared without reason | | .75*** | | .17*** | .57*** | .10 | .70*** | | -.0*** | | .70*** | .04 | .32** | -.06 |
| 1 | Hard to wind down | | | .09* | .06 | -.33*** | .41*** | .09* | | | .02 | .16** | -.02 | -.35*** | -.18 |
| 6 | Tended to over-react | | | .69*** | .14** | .36*** | .32*** | .68*** | | | .27** | .64*** | .01 | .05 | .32*** |
| 8 | Using nervous energy | | | .75*** | .08 | .49*** | .33*** | .74*** | | | -.01 | .72*** | -.04 | .13* | .18** |
| 11 | Getting agitated | | | .69*** | .20*** | .29*** | .31*** | .68*** | | | .08 | .63*** | .06 | .01 | .28*** |
| 12 | Difficult to relax | | | .70*** | .20*** | .38*** | .24*** | .71*** | | | -.19 | .66*** | .06 | .11* | .14** |
| 14 | Intolerant of restrictions | | | .52*** | .30*** | .00 | .30*** | .51*** | | | .23** | .47*** | .15** | -.19** | .29** |
| 18 | Felt rather touchy | | | .63*** | .18*** | .21*** | .36*** | .62*** | | | .16* | .60*** | .05 | -.06 | .22** |

CFA = confirmatory factor analysis; ESEM = exploratory structural equation model; BESEM = bi-factor exploratory structural equation model.

*p < .05.

**p < .01.

the stress factor. With the exception of one item (Item 1), the loadings for the other items were all salient and positive. Four depression items (Items 3, 5, 16 and 21) loaded significantly on the stress factor. Five anxiety items (Items 2, 4, 7, 15 and 19) loaded significantly on stress factor, with three of them (Items 2, 4 and 19) being salient.

Therefore, taken as a whole, the considerable significant cross-loadings (many being also salient) as well as the negative loading of items on their targeted factors indicate that all the specific factors were not well defined. Psychometrically, the substantial overlap in variances across the items belonging to the different factors raises the possibility that a higher-order factor model (such as BCFA) is be needed to better model the variances in the DASS-21 items.

## BCFA versus BESEM models

**Factor loadings and cross-loadings.**   Table 3 shows the factor loadings for the BCFA model. For the BCFA model, all 21 items loaded significantly on the general factor, with 20 loading being salient. The exception was Item 1. All seven depression items loaded significantly and positively on the specific depression factor, with six items having salient loadings (the exception being Item 5). All seven anxiety items loaded negatively on the specific anxiety factor, with six items (Items 4, 7, 9, 15, 19 and 20) being significant, but not salient. Only three stress items (Items 6, 14 and 18) loaded significantly on the specific stress factor, and none of them were salient. Also, two items (Items 8 and 12) loaded negatively. Taken together, the pattern of negative loadings, and the strength of loadings can be taken to indicate that although the BCFA model showed good global fit, and the general factor and the specific depression factors were at least adequately defined, the specific factors for stress and anxiety in this model were not well defined.

Table 3 also includes the factor loadings for the BESEM with target rotation model. For this model, all 21 items loaded significantly on the general factor, with 20 loading being salient. The exception was Item 1. All seven depression items loaded significantly on the specific depression factor, with six items having salient loadings. The exception was Item 5. Two anxiety items (Items 14 and 19) loaded significantly and negatively, and one anxiety item (Item 9) loaded significantly and positively on the specific depression factor. Also, one stress item (Item 15) loaded significantly and positivity on the specific depression factor. Thus, the specific depression factor was at best moderately well defined. With the exception of two anxiety items (Items 4 and 19), the other five anxiety items loaded significantly on the specific anxiety factor. Three items (Items 9, 15 and 20) loaded positively and saliently, and the other one item (Item 2) loaded negatively and saliently. Three depression items (Items 3, 13, and 16) loaded significantly (but not saliently) on the specific anxiety factor, with two of them (Items 3 and 16) being negative. With the exception of one stress item (Item 6), the remaining six stress items loaded significantly and positively on the specific anxiety factor. The loading for Item 1 was salient and negative, and the loading for Item 14 was negative. Thus the specific anxiety factor was poorly defined. With the exception of one stress item (Item 1), the other six stress items loaded significantly and positively on the specific stress factor, with only one loading (Item 6) being salient. With the exception of three depression items (Items 13, 16 and 21), the other four anxiety items loaded significantly (but not saliently) on the specific stress factor, with one of them (Item 3) being negative. None of the anxiety items loaded significantly on the specific stress factor. Four of these items loaded negatively. Therefore, the specific stress factor can also be considered to be poorly defined.

Overall, like the ESEM with target rotation model, the BESEM with target rotation model also indicated poorly defined stress and anxiety factors. Additionally, several items in the BESEM with target rotation model lacked significant and salient positive loading on the

targeted factors, or loaded significantly and saliently on non-target factors. Overall, with the exception of Item 5, none of the other depression items fell into any of these categories. For the anxiety items, Items 2, 4, 7, 19 and 20 fell into one or more of these categories. For the stress items, Items 1, 8, 11, 12, 14 and 18 fell into one or more of these categories.

## Reliabilities of the factors in all the DASS-21 models tested

For the three-factor CFA model, the ECV for the depression, anxiety, and stress were .43, .30, and .26, respectively. The categorical omega reliability coefficients (95% percentile boot-strapped confidence intervals) for the factors this model were depression = .88 (CI = .87–90), anxiety = .78 (CI = .75–81), and stress = .75 (CI—.71–77). These values can be considered at least acceptable ($\geq 0.70$). For the BCFA model, the ECV for the general factor, and the specific factors for depression, anxiety, and stress were .81, .16, .00, and .02 respectively. The $\omega_h$ value for the general factor was .92, and the $\omega_s$ values for the specific factors for depression, anxiety, and stress were .26, .22, and .22, respectively. Thus although the reliability of the general factor was good, they were low for all three specific factors. Overall therefore, across these models, the CFA model showed the better reliabilities for its factors.

## Validities of the factors in all the DASS-21 models tested

The standardized path coefficients (probability) for prediction of IA and HI symptoms by gender, age, general distress, depression, anxiety, and stress are shown in Table 4. In the CFA model, IA was associated with stress and anxiety, whereas HI was associated with stress and depression. The differential associations for the DASS-21 factors with IA and H can be interpreted as supportive of the external validity of the DASS-21 factors, especially depression and anxiety. For the ESEM with target rotation model, IA was associated with stress, and HI was associated with stress and anxiety. Thus there was support for the externality validity of only the anxiety factor. For the BCFA model, IA and HI were both associated with the distress, depression, and anxiety factors, HI ws also associated with the stress factors. These findings indicate support for the external validity of only the distress factor and specific stress factor. For the BESEM with target rotation model, the findings indicate that both IA and HI were associated positively with distress, and negatively with anxiety. The findings can be interpreted to indicate support for the external validity of the general ADHD factor, but not the specific

**Table 4. Standardized beta coefficients for the predictions of the DASS scale (depression, anxiety, and stress) scores by the factors in the bi-factor ESEM model, with gender and age as covariates.**

|  | CFA | | ESEM | | BCFA | | BESEM | |
|---|---|---|---|---|---|---|---|---|
|  | **IA** | **HI** | **IA** | **HI** | **IA** | **HI** | **IA** | **HI** |
| Gender | 0.12** | 0.08* | 0.12** | 0.08* | 0,12 | 0.08 | 0.12** | 0.08* |
| Age | -0.12** | -0.10** | -0.12** | -0.10** | -0.12 | -0.10 | -0.12** | -0.10** |
| General/Distress | - | - | - | - | 0.79*** | 0.86*** | 0.60*** | 0.59*** |
| Depression | 0.06 | 0.18** | 0.10 | -0.05 | -0.33*** | -0.56*** | -0.07 | -0.17 |
| Anxiety | 0.15** | 0.06 | 0.06 | 0.18** | -0.51** | -0.64** | -0.17** | -0.12* |
| Stress | 0.57*** | 0.57*** | 0.52*** | 0.52*** | 0.96 | 1.28* | 0.08 | 0.14 |

IA = Inattention; HI = Hyperactivity/Impulsivity BESEM = bi-factor exploratory structural equation model.

***$p < .001$

**$p < .01$

*$p < .05$

factors for depression, anxiety, and stress. Taken together, the findings can be interpreted as providing most support for the external validity of the factors in the CFA model.

## Discussion

### Selection of the optimum DASS-21 model

The present study examined the structure of the DASS-21 items among adults from the general community using CFA, BCFA, ESEM with target rotation, and BESEM with target rotation approaches. The three group/specific factors in all models were depression, anxiety, and stress. A general factor was also included in the bi-factor models. In terms of global fit values, all the models showed good fit. However, given that for the ESEM with target rotation, BCFA, and BESEM with target rotation models: (i) one or more of their specific/group factors were poorly defined; (ii) the specific factors for depression, anxiety and stress in the BESEM with target rotation model had low reliabilities; (iii) and the lack of support for their external validities, the ESEM with target rotation, BCFA, and BESEM with target rotation models were deemed as unacceptable. Because the CFA model had good global fit; (ii) loadings that were significant and salient; (iii) factors with acceptable reliabilities and validities; and (iv) the most parsimonious of all the models tested, this model was selected as the preferred model.

The findings here are in part consistent with past studies that have compared the factor structure of the DASS-21 using CFA, BCFA, and ESEM with target rotation and BESEM with target rotation approaches [6,7]. Johnson [6] found that although the ESEM with target rotation model showed the best fit (compared to CFA and BCFA model), the anxiety factor was poorly defined, with several items loading significantly onto the depression or stress factors. They also found that although the BESEM with target rotation model with specific factors for depression, anxiety and stress and a general factor showed slightly better fit than the ESEM model, the improvement was not sufficient to justify the loss of parsimony. Kyriazos [7] found that although the CFA, ESEM with target rotation, and BCFA models showed good fit, the factor loadings in the BCFA and ESEM with target rotation models were unsatisfactory (low and/ or negative item loadings on targeted factors). Consequently, as with the findings of the present study, they concluded most support for the CFA model. In addition to the comparability of the findings here to previous findings, the present study's findings also extend the extant findings in this area because unlike the two previous studies, the present study also examined the factor structure of DASS-21 in terms of the BESEM with target rotation model in a community-based sample. Despite being a more advanced model for testing the factor structure of a measure, the DASS-21 BESEM model was also not satisfactory because the anxiety and stress factors in this model were poorly defined, and there was a lack of support for the reliabilities and validities of its specific factors. However, it is worthy of note that there was support for the general factor in this model.

Although, the present study adopted the CFA 3-factor model as the preferred model for DASS-21, this model is not without limitations. First, the findings in the study did provide support for modeling a higher order factor because the correlations of latent factors in the CFA model were higher than the corresponding correlations in the ESEM with target rotation model. Second, because the ESEM with target rotation model showed considerable cross-loadings, it follows that there is a need to include a general factor to capture the shared variances in the DASS-21 items. Despite the fact that the BCFA and BESEM with target rotation modeled a general factor, both models showed poorly defined anxiety and stress factors. Consequently, it is possible that the description and/or content of the items the DASS-21 may not be appropriate. Given that the BESEM models provides a more advanced method to model the variances among the DASS-21 items, the factor loadings of the items in the specific factors this model

can be used to identify problematic items. In this respect, items that (i) lack significant and salient positive loading on the targeted factors, (ii) do not load significantly on the targeted factors, or (iii) loaded significantly and saliently on non-target factors, can be considered as problematic. The findings here showed that with the exception of Item 5, none of the other depression items fell into any of these categories. For the anxiety items, Items 2, 4, 7, 19 and 20 fell into one or more of these categories; and for the stress items, Items 1, 8, 11, 12, 14 and 18 fell into one or more of these categories. Therefore, Items 1, 2, 4, 5, 7, 8, 11, 12, 14, 19, 18, and 20 in the DASS-21 are probably the items that may need to be reviewed for inclusion in the DASS-21. If the suggested items are removed, it would mean a shorter questionnaire with just six depression items, three anxiety items, and one stress item. The depression items would include Items 3 ("expressing positive feelings"), 10 ("nothing to look forward to"), 13 ("felt down-hearted"), 16 ("unable to become enthusiastic"), 17 ("not worth much"), and 21 ("life was meaningless"); and the anxiety items could include Items 9 ("worry about embarrassing self"), 15 ("felt close to panic"), and 20 (felt scared without reason"). Therefore, as intended by the developers of the DASS-21 [3], and in line with the tripartite model [39,40], the remaining depression items will still be assessing dysphoria, low self-esteem, and lack of incentive; and the remaining anxiety items will still be assessing subjective responses to anxiety and fear. The one stress item (Item 6) will be assessing "tended to over-react", that is, as intended, negative affectivity response reflecting nervous tension and irritability.

## Limitations and conclusions

The current research conclusions may require to be perceived in the light of some restrictions. Specifically, provided that the ethic approval did not enable the gathering of participants' information before inviting them to enroll in the study, no information is accessible about any potential communalities between those who were exposed to the study invitation but decided not to contribute. Thus, it is likely that the final sample examined in the study may not be representative of the general population. Furthermore, as the current research involved a community sample, it may be questionable how the results may apply to clinical samples. Third, because the DASS-21 is a self-report questionnaire, it is possible the ratings may have been influence by the method used to collect them, thereby subjecting participants to common method variance effect. Fourth, the conclusions made in this study are based on a single study. To explore the robustness of the study's findings, they need to be replicated. Given these limitations, the results of the study may be viewed as preliminary. Despite these potential restrictions, it is argued that the present study findings do support more studies in this area, accounting for the limitations identified here.

## Supporting information

**S1 Data.**
(SAV)

## Author Contributions

**Conceptualization:** Rapson Gomez, Vasileios Stavropoulos, Mark D. Griffiths.

**Data curation:** Rapson Gomez, Vasileios Stavropoulos.

**Formal analysis:** Rapson Gomez.

**Investigation:** Rapson Gomez.

**Methodology:** Rapson Gomez, Vasileios Stavropoulos.

**Supervision:** Mark D. Griffiths.

**Writing – original draft:** Rapson Gomez, Vasileios Stavropoulos.

**Writing – review & editing:** Rapson Gomez, Vasileios Stavropoulos, Mark D. Griffiths.

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
