## [Decision Letter · Decision Letter 0]

27 Jan 2020

PONE-D-19-32096

Confirmatory Factor Analysis and Exploratory Structural Equation Modeling of the Factor Structure of the Depression Anxiety Stress Scales-21

PLOS ONE

Dear Dr. Stavropoulos,

Thank you for submitting your manuscript to PLOS ONE. After careful consideration, we feel that it has merit but does not fully meet PLOS ONE’s publication criteria as it currently stands. Therefore, we invite you to submit a revised version of the manuscript that addresses the points raised during the review process.

We would appreciate receiving your revised manuscript by Mar 12 2020 11:59PM. To enhance the reproducibility of your results, we recommend that if applicable you deposit your laboratory protocols in protocols.io, where a protocol can be assigned its own identifier (DOI) such that it can be cited independently in the future. For instructions see: http://journals.plos.org/plosone/s/submission-guidelines#loc-laboratory-protocols

We look forward to receiving your revised manuscript.

Kind regards,

Manuel Fernández-Alcántara, Ph.D.

Academic Editor

PLOS ONE

Journal Requirements:

2. We note that some of the partcipants in your study were under the age of 18. Please state in your methods section whether you obtained consent from parents or guardians of the minors included in the study or whether the research ethics committee or IRB approved the lack of parent or guardian consent.

Reviewers' comments:

Reviewer's Responses to Questions

**Comments to the Author**

1. Is the manuscript technically sound, and do the data support the conclusions?

Reviewer #1: Partly

Reviewer #2: Yes

2. Has the statistical analysis been performed appropriately and rigorously? 

Reviewer #1: Yes

Reviewer #2: Yes

3. Have the authors made all data underlying the findings in their manuscript fully available?

Reviewer #1: No

Reviewer #2: Yes

4. Is the manuscript presented in an intelligible fashion and written in standard English?

Reviewer #1: Yes

Reviewer #2: Yes

5. Review Comments to the Author

Reviewer #1: Overall, this was a robust and interesting evaluation of the measurement structure of the DASS-21. I have only minor comments for clarification.

The authors state several times that they are the first to test a bifactor ESEM of the DASS-21, but it should be noted that Johnson et al. (2016) also tested two Bifactor ESEM Models, including the BESEM Model tested in the present paper. See the following from the Methods section in Johnson et al. (2016):

Five established DASS-21 factor structures were fit first in CFA, and then in ESEM with ‘Target’ rotation:

1. 1-factor Model

2. 2-factor Model – Oblique model, two correlated factors representing physiological hyperarousal and generalised negativity, Duffy et al., (2005)

3. 3-factor Model – Oblique model, three correlated factors representing depression, anxiety, and stress, Lovibond and Lovibond (1995)

4. Bifactor Model A – Nested model, 3 independent factors representing depression, anxiety, and stress and a general negative affect factor, Henry and Crawford (2005)

5. Bifactor Model B – Nested model, 2 independent factors representing depression and stress and a general negative affect factor, Tully, Zajac, and Venning (2009)

A note on the wording: there is extensive reference to the method used as simply ‘ESEM’, but ESEM is a general framework for using EFA within structural models and does not indicate the type of EFA rotation used. The authors should note that the analyses used ‘Target’ rotation.

The calculation of the Omega coefficient needs clarification. When using the WLSMV estimator, the Omega statistic should not be calculated in the same way as is done with the analysis of continuous items (Yang & Xia, 2019). A method for estimating coefficient omega for ordinal items has been proposed by Green and Yang (2009) and should be used instead.

The authors should clarify the extent of missingness present in the data. WLSMV is a limited-information estimator and handles missingness using pairwise deletion. This approach may not be suitable for samples with large proportions of missingness.

Minor correction: the estimator is not named ‘Weighted least square mean and variance adjusted chi-square’, just ‘Weighted least square mean and variance adjusted’, as both the standard errors and the chi-square statistic are adjusted.

While the authors provide fit indices for each model, they do not provide the model comparison statistics (especially the Δχ2). While the authors mention that ‘The models did not differ from each other because the ΔCFI and ΔRMSEA values between all model pairs were less than 0.01, and 0.015 respectively.’, this is not the case. The ΔCFI was greater than 0.01 for the comparison between the CFA 3-F and all models except the BCFA 3-s-F.

The authors state in the discussion ‘Consequently, it is possible that there may be another non-modeled higher order factor needed for the DASS-21.’ This is a simple hypothesis to test, as the authors could add a higher-order 3-factor model (i.e. the 3 factors as indicators of another latent variable).

The authors mention several times that BESEM ‘provides the most advanced method to model the variances among the DASS-21 items…’. However, this title would likely go to Bayesian SEM, which can estimate both cross-loadings and residual covariances (whereas ESEM & BESEM can only estimate cross-loadings).

The authors’ conclusion that ‘…the total scores of the depression, anxiety, and stress scales could be used as measures of these constructs’ is incongruent with the findings of multiple cross-loadings in the ESEM models. Using sum scores for scale totals assumes that the scale is measured only by the items being summed, and that severity on these items is indicative only of severity on the construct of interest (measurement error aside). By identifying several cross-loading items, the authors have indicated that these assumptions do not hold, and that total scores may not be reliable.

Green, S. B., & Yang, Y. (2009). Reliability of Summed Item Scores Using Structural Equation Modeling: An Alternative to Coefficient Alpha. Psychometrika, 74(1), 155-167. doi:10.1007/s11336-008-9099-3

Yang, Y., & Xia, Y. (2019). Categorical Omega With Small Sample Sizes via Bayesian Estimation: An Alternative to Frequentist Estimators. Educational and Psychological Measurement, 79(1), 19-39. doi:10.1177/0013164417752008

Reviewer #2: The subject is of interest and the manuscript has been performed with quality methodological. However, some aspects should be worked on, namely the quality of the writing and the content to be expressed in each part of the manuscript.

Introduction

- The authors don't have provided literature to justify the need for testing the DASS-21. The beginning "The DASS-21 has been derived from...." is not an appropriate way to start a manuscript. It should include a brief introduction of the topic to be addressed, of the importance of this scale... And not start with a psychometric approach. In this same line, concepts such as CFA and ESSEM are first introduced to be explained in the final part of the introduction; this does not facilitate the understanding of the subject.

- The authors make multiple statements throughout the translation that should be referenced, giving rise to confusion because they seem to have given it.

- The introduction is excessively long and confusing in its presentation. It needs to be worked on in-depth. It is not properly structured, with ideas repeated throughout the text that do not facilitate understanding and a vision of progression and organization.

- Acronyms must be specified the first time they appear. Even the name of a scale.

- On page 6, refer to "minor misspecifications", please clarify such general expressions.

Method:

- There is no need to indicate age by gender.

- Better express the statistical results (M = , SD = )

- The expression (t[736]=0.05, ns), is incorrectly expressed. Please include the statistic t correctly and the p.

- It is unnecessary to include the descriptive for the DASS items.

- It would be convenient to include other types of sociodemographics in the table, such as the year, the gender, the nationality, socioeconomics level.

- The last paragraph of the participants is for discussion, not methods.

- More detailed information needs to be provided on sample and recruitment, procedure.

- It would be recommended to reduce the analysis data section.

- It has not been included which type of statistical tests have been carried out in the validity of the tool.

Results

- Summarize the results in the text if they are set out in the tables.

- It is not necessary to explain in detail in the text the factorial loads of each of the items. Summarize and specify the different and/or relevant results. Also, organize it in a detailed way by the factor in each of the models. As it is now, it is confusing and disorganized.

- The results of the statistical tests of the validation of the instrument are not properly expressed.

- The sentence: "Overall, these findings indicate support for the external validity..." is not methodologically correct. This is the internal validity of the instrument, not external validity (generalization of the results to the general population).

- The paragraph where the authors explain the preferred model is more appropriate in the discussion section. In the results section, the results are not discussed, only a statement of the results should be made.

- In the tables, all abbreviations should be specified in notes

Discussion

- The discussion must be worked on, providing a greater comparison with other studies and interpretation of results. They should avoid re-summarizing the results, putting paragraphs where not a single reference appears.

- The conclusion of the discussion section should not include interpretation of the results.

6. PLOS authors have the option to publish the peer review history of their article (what does this mean?). If published, this will include your full peer review and any attached files.

Reviewer #1: Yes: Andrew R Johnson

Reviewer #2: No

---

## [Author Response · Author response to Decision Letter 0]

23 Feb 2020

04th Feburary 2020

The Editor

PLOS ONE

Re: Revision of Manuscript PONE-D-19-32096

Confirmatory Factor Analysis and Exploratory Structural Equation Modeling of the Factor Structure of the Depression Anxiety Stress Scales-21

Dear Editor,

Thank you very much for arranging the review of the above mentioned paper that was submitted to PLOS ONE. We also like the reviewers for their valuable comments. These have been extremely useful for the revision of the paper. In the new revised paper we have considered these comments, and we believe that we have responded to all the revisions suggested. Our revisions are listed below, and are highlighted in yellow and green in the revised paper

Responses to Reviewer #1: 

Overall, this was a robust and interesting evaluation of the measurement structure of the DASS-21. I have only minor comments for clarification.

Comment: The authors state several times that they are the first to test a bifactor ESEM of the DASS-21, but it should be noted that Johnson et al. (2016) also tested two Bifactor ESEM Models, including the BESEM Model tested in the present paper. See the following from the Methods section in Johnson et al. (2016):

Five established DASS-21 factor structures were fit first in CFA, and then in ESEM with ‘Target’ rotation:

1. 1-factor Model

2. 2-factor Model – Oblique model, two correlated factors representing physiological hyperarousal and generalised negativity, Duffy et al., (2005)

3. 3-factor Model – Oblique model, three correlated factors representing depression, anxiety, and stress, Lovibond and Lovibond (1995)

4. Bifactor Model A – Nested model, 3 independent factors representing depression, anxiety, and stress and a general negative affect factor, Henry and Crawford (2005)

5. Bifactor Model B – Nested model, 2 independent factors representing depression and stress and a general negative affect factor, Tully, Zajac, and Venning (2009)

Response. We have removed all instances where we have claimed (previously) that we are the first to test a bifactor ESEM of the DASS-21. Related to this, we have cited and discussed the work of Johnson et al. (2016), in particular their exploration and findings related to the bifactor ESEM model (p. 10, yellow highlight I n para 1).

Comment: A note on the wording: there is extensive reference to the method used as simply ‘ESEM’, but ESEM is a general framework for using EFA within structural models and does not indicate the type of EFA rotation used. The authors should note that the analyses used ‘Target’ rotation.

Response. As suggested we no longer refer to ESEM, but to ESEM (and also BESEM) with target rotation (highlighted in yellow throughout the paper).

Comment: The calculation of the Omega coefficient needs clarification. When using the WLSMV estimator, the Omega statistic should not be calculated in the same way as is done with the analysis of continuous items (Yang & Xia, 2019). A method for estimating coefficient omega for ordinal items has been proposed by Green and Yang (2009) and should be used instead.

Response. We have not made any change in this respect, for the reasons given in p. 15 (yellow highlight).

Comment: The authors should clarify the extent of missingness present in the data. WLSMV is a limited-information estimator and handles missingness using pairwise deletion. This approach may not be suitable for samples with large proportions of missingness.

Response. We covered this in p. 15 (yellow highlight).

Comment Minor correction: the estimator is not named ‘Weighted least square mean and variance adjusted chi-square’, just ‘Weighted least square mean and variance adjusted’, as both the standard errors and the chi-square statistic are adjusted.

Response. Corrected (p. 13, yellow highlight).

Comment: While the authors provide fit indices for each model, they do not provide the model comparison statistics (especially the Δχ2). While the authors mention that ‘The models did not differ from each other because the ΔCFI and ΔRMSEA values between all model pairs were less than 0.01, and 0.015 respectively.’, this is not the case. The ΔCFI was greater than 0.01 for the comparison between the CFA 3-F and all models except the BCFA 3-s-F.

Response. This mistake has been correct (p. 16, yellow highlight)

Comment: The authors state in the discussion ‘Consequently, it is possible that there may be another non-modeled higher order factor needed for the DASS-21.’ This is a simple hypothesis to test, as the authors could add a higher-order 3-factor model (i.e. the 3 factors as indicators of another latent variable).

Response. We now consider that based on our findings such an hypothesis is inappropriate, and have remove this statement. Thus the computation of the model suggested is irrelevant. 

Comment: The authors mention several times that BESEM ‘provides the most advanced method to model the variances among the DASS-21 items…’. However, this title would likely go to Bayesian SEM, which can estimate both cross-loadings and residual covariances (whereas ESEM& BESEM can only estimate cross-loadings).

Response. We no longer refer to the BESEM approach as the most advanced method to model the variances among the DASS-21 items…’. Instead we say it is a “could be potentially a more sophisticated and arguably more advanced approach for studying the factor structure of the DASS-21” (p. 10, lines 11 & 12).

Comment: The authors’ conclusion that ‘…the total scores of the depression, anxiety, and stress scales could be used as measures of these constructs’ is incongruent with the findings of multiple cross-loadings in the ESEM models. Using sum scores for scale totals assumes that the scale is measured only by the items being summed, and that severity on these items is indicative only of severity on the construct of interest (measurement error aside). By identifying several cross-loading items, the authors have indicated that these assumptions do not hold, and that total scores may not be reliable.

Response. Given the suggestion by Reviewer 2 that the “conclusion of the discussion section should not include interpretation of the results” we have removed the paragraph that contains the information referred to by this reviewer. Thus this point in not made in the current version of the paper. 

Responses to Reviewer #2: 

The subject is of interest and the manuscript has been performed with quality methodological. However, some aspects should be worked on, namely the quality of the writing and the content to be expressed in each part of the manuscript.

Comment: Introduction

- The authors don't have provided literature to justify the need for testing the DASS-21. The beginning "The DASS-21 has been derived from...." is not an appropriate way to start a manuscript. It should include a brief introduction of the topic to be addressed, of the importance of this scale... And not start with a psychometric approach. In this same line, concepts such as CFA and ESSEM are first introduced to be explained in the final part of the introduction; this does not facilitate the understanding of the subject.

Response: Justification for the study is now provided (p. 10, green highlights). We begin the paper now with a a brief introduction of the topic to be addressed (p. 3 para 1, green highlights). We have revised the paper such that concepts such as CFA and ESSEM are not introduced first without explanation.

Comment: The authors make multiple statements throughout the translation that should be referenced, giving rise to confusion because they seem to have given it.

Response. Done throughout.

Comment: The introduction is excessively long and confusing in its presentation. It needs to be worked on in-depth. It is not properly structured, with ideas repeated throughout the text that do not facilitate understanding and a vision of progression and organization.

Response. We have reorganized the introduction, providing better structure, and removing repetitions. 

Comment: Acronyms must be specified the first time they appear. Even the name of a scale.

Response. This has been conformed to.

Comment: On page 6, refer to "minor misspecifications", please clarify such general expressions.

Response. This has been clarified (p. 5, green highlight). 

Method:

Comment - There is no need to indicate age by gender.

- Better express the statistical results (M = , SD = )

- The expression (t[736]=0.05, ns), is incorrectly expressed. Please include the statistic t correctly and the p.

Response. All the above have been revised (see “Participants” section, p. 11)

Comment: It is unnecessary to include the descriptive for the DASS items.

Response: Removed (see Table 1)

Comment: It would be convenient to include other types of sociodemographics in the table, such as the year, the gender, the nationality, socioeconomics level.

Response: As the information on age and gender have been provided in the text, we hve not provided this in the table. There is no information on nationality and socioeconomics level. 

Comment: The last paragraph of the participants is for discussion, not methods.

Response: It still remain under participants, we fell that it provides information on the participants that is better at this point of the paper. 

Comment: More detailed information needs to be provided on sample and recruitment, procedure.

Response: Additional information is now provided (p. 13, green highlight)

Comment: It would be recommended to reduce the analysis data section.

Response: We have done so.

Comment: It has not been included which type of statistical tests have been carried out in the validity of the tool.

Response: We have done so (p. 15, green highlight).

Results

Comment: Summarize the results in the text if they are set out in the tables.

Response: We have done so.

Comment: It is not necessary to explain in detail in the text the factorial loads of each of the items. Summarize and specify the different and/or relevant results. Also, organize it in a detailed way by the factor in each of the models. As it is now, it is confusing and disorganized.

Response: We have not made much changes in this respect. Our intention in the results section is to show how well the factors are defined by the individual items, and latter to discuss which items are problematic. Dscussion of these will make little sense if details of individual items are not covered in the results section. 

Comment: The results of the statistical tests of the validation of the instrument are not properly expressed.

Response: We have rewritten this section (pp. 20 & 21, green highlight).

Comment: The sentence: "Overall, these findings indicate support for the external validity..." is not methodologically correct. This is the internal validity of the instrument, not external validity (generalization of the results to the general population).

Response. We think that examining the relations of the DASs-21 factors with IA and HI test the external (and not internal) validity of the DASS-21 factors. 

- The paragraph where the authors explain the preferred model is more appropriate in the discussion section. In the results section, the results are not discussed, only a statement of the results should be made.

Response: This is now only in the discussion. 

Comment: In the tables, all abbreviations should be specified in notes

Response: Done

Discussion

Comment: The discussion must be worked on, providing a greater comparison with other studies and interpretation of results. They should avoid re-summarizing the results, putting paragraphs where not a single reference appears.

Response: In our discussion, we have providing a greater comparison with other studies and made interpretation of results. Although there are paragraphs without reference, we consider these paragraphs as essential to explaining our results. 

Comment: The conclusion of the discussion section should not include interpretation of the results.

Response: the interpretation of results has been removed. 

As you will notice we have addressed all the comments and suggestions made by the reviewer. We hope that the changes meet with your satisfaction and we look forward to hearing from you soon.

Thank you. 

Yours sincerely,

Authors

---

## [Decision Letter · Decision Letter 1]

20 Mar 2020

PONE-D-19-32096R1

Confirmatory Factor Analysis and Exploratory Structural Equation Modeling of the Factor Structure of the Depression Anxiety Stress Scales-21

PLOS ONE

Dear Dr. Stavropoulos,

Thank you for submitting your manuscript to PLOS ONE. After careful consideration, we feel that it has merit but does not fully meet PLOS ONE’s publication criteria as it currently stands. Therefore, we invite you to submit a revised version of the manuscript that addresses the points raised during the review process.

Thank you for submitting your research to PlosOne. Before final acceptance of the manuscript please address the minor revisions suggested by Reviewer 1. With this clarification the manuscript will be ready for publication. Congratulations for your interesting and applied work.

We would appreciate receiving your revised manuscript by May 04 2020 11:59PM. To enhance the reproducibility of your results, we recommend that if applicable you deposit your laboratory protocols in protocols.io, where a protocol can be assigned its own identifier (DOI) such that it can be cited independently in the future. For instructions see: http://journals.plos.org/plosone/s/submission-guidelines#loc-laboratory-protocols

We look forward to receiving your revised manuscript.

Kind regards,

Manuel Fernández-Alcántara, Ph.D.

Academic Editor

PLOS ONE

Reviewers' comments:

Reviewer's Responses to Questions

**Comments to the Author**

1. If the authors have adequately addressed your comments raised in a previous round of review and you feel that this manuscript is now acceptable for publication, you may indicate that here to bypass the “Comments to the Author” section, enter your conflict of interest statement in the “Confidential to Editor” section, and submit your "Accept" recommendation.

Reviewer #1: (No Response)

Reviewer #2: All comments have been addressed

2. Is the manuscript technically sound, and do the data support the conclusions?

Reviewer #1: Yes

Reviewer #2: Yes

3. Has the statistical analysis been performed appropriately and rigorously? 

Reviewer #1: Yes

Reviewer #2: Yes

4. Have the authors made all data underlying the findings in their manuscript fully available?

Reviewer #1: Yes

Reviewer #2: (No Response)

5. Is the manuscript presented in an intelligible fashion and written in standard English?

Reviewer #1: Yes

Reviewer #2: Yes

6. Review Comments to the Author

Reviewer #1: In their revision, the authors stated that:

"It may be worth noting that Yang and Green (2015) have argued that when

using the WLSMV estimator, the omega should not be calculated in the same way as is done

with the analysis of continuous items (Yang & Xia, 2019). This is especially so if the sample size

is less than 500 as omega values would be underestimated. Instead, Bayesian estimation methods

have been proposed (Yang & Xia, 2019). However, as our sample size was well above 500 (N =

738), we used the usual method applied to continuous items for computing omega values."

However, these are not the conclusions of the studies that they are citing. The referenced studies show that *categorical* Omega (i.e. the correct treatment of omega with WLSMV estimation) can be biased in small sample sizes, not that the standard (continuous) version is biased. In fact, the referenced studies do not predicate the appropriateness of the continuous method on sample size at all:

"However, x and factor scores have different metrics and their relationship is no longer linear. Describing the relation of the observed scores to the true scores using the Pearson correlation (or the square of the correlation) is thus inappropriate (Lord & Novick, 1968). Consequently, Definition 2 and Definition 3 are not appropriate for defining a reliability coefficient for the scale scores." (Yang & Xia, 2018, p. 21).

The authors need to use categorical Omega here, otherwise the assumptions on which they estimate reliability (continuous observations) are different from the assumptions used to estimate the models (ordinal observations). If unsure, the simplest way to achieve this is using the ci.reliability() function in the MBESS R package.

Reviewer #2: (No Response)

7. PLOS authors have the option to publish the peer review history of their article (what does this mean?). If published, this will include your full peer review and any attached files.

Reviewer #1: Yes: Andrew R. Johnson

Reviewer #2: No

---

## [Author Response · Author response to Decision Letter 1]

15 May 2020

15h May 2020

The Editor

PLOS ONE

Re: Revision of Manuscript PONE-D-19-32096

Confirmatory Factor Analysis and Exploratory Structural Equation Modeling of the Factor Structure of the Depression Anxiety Stress Scales-21

Dear Editor,

Thank you very much for arranging the second review of the above mentioned paper that was submitted to PLOS ONE. We also like the reviewer 2 (who did not request any additional modifications) and reviewer 1 for recommending the calculation of the categorical omega via the R-MBESS ci-reliability function. In the new revised paper we have considered this comment raised by reviewer 1. We believe that we have now responded to this final revision suggested. The revision(s) is listed below, and is highlighted in yellow in this second revised version of the paper.

Responses to Reviewer #1: 

Comment: In their revision, the authors stated that:

"It may be worth noting that Yang and Green (2015) have argued that when

using the WLSMV estimator, the omega should not be calculated in the same way as is done

with the analysis of continuous items (Yang & Xia, 2019). This is especially so if the sample size

is less than 500 as omega values would be underestimated. Instead, Bayesian estimation methods

have been proposed (Yang & Xia, 2019). However, as our sample size was well above 500 (N =

738), we used the usual method applied to continuous items for computing omega values."

However, these are not the conclusions of the studies that they are citing. The referenced studies show that *categorical* Omega (i.e. the correct treatment of omega with WLSMV estimation) can be biased in small sample sizes, not that the standard (continuous) version is biased. In fact, the referenced studies do not predicate the appropriateness of the continuous method on sample size at all:

"However, x and factor scores have different metrics and their relationship is no longer linear. Describing the relation of the observed scores to the true scores using the Pearson correlation (or the square of the correlation) is thus inappropriate (Lord & Novick, 1968). Consequently, Definition 2 and Definition 3 are not appropriate for defining a reliability coefficient for the scale scores." (Yang & Xia, 2018, p. 21).

The authors need to use categorical Omega here, otherwise the assumptions on which they estimate reliability (continuous observations) are different from the assumptions used to estimate the models (ordinal observations). If unsure, the simplest way to achieve this is using the ci.reliability() function in the MBESS R package.

Response. Thank you for your comment. The relevant part of the text has now been modified. Furthermore, we have computed and reported categorical Omega using the the ci.reliability function in the MBESS R package and perc interval type (see p.17, yellow highlight; and p. 22, yellow highlight). As we have been using M-Plus explicitly until now, your review point has provided us the opportunity to get more familiar with R and we are honestly grateful to you for that. 

Responses to Reviewer #2: 

Comment: No additional comments.

Response: Thank you for the time you devoted in the consideration of our manuscript and the effort you made to help us improve it.

As you will notice we have addressed all the comments and suggestions made by the reviewer. We hope that the changes meet with your satisfaction and we look forward to hearing from you soon.

Thank you. 

Yours sincerely,

Authors

---

## [Editor Report · Decision Letter 2]

18 May 2020

Confirmatory Factor Analysis and Exploratory Structural Equation Modeling of the Factor Structure of the Depression Anxiety Stress Scales-21

PONE-D-19-32096R2

Dear Dr. Stavropoulos,

We are pleased to inform you that your manuscript has been judged scientifically suitable for publication and will be formally accepted for publication once it complies with all outstanding technical requirements.

With kind regards,

Manuel Fernández-Alcántara, Ph.D.

Academic Editor

PLOS ONE
---

## [Editor Report · Acceptance letter]

22 May 2020

PONE-D-19-32096R2 

Confirmatory Factor Analysis and Exploratory Structural Equation Modelling of the Factor Structure of the Depression Anxiety Stress Scales-21 

Dear Dr. Stavropoulos:

I am pleased to inform you that your manuscript has been deemed suitable for publication in PLOS ONE. Congratulations! Your manuscript is now with our production department. 

With kind regards,

on behalf of

Dr. Manuel Fernández-Alcántara 

Academic Editor

PLOS ONE